# Beyond Worst-case: A Probabilistic Analysis of Affine Policies in Dynamic Optimization

**Omar El Housni**
IEOR Department
Columbia University
oe2148@columbia.edu

**Vineet Goyal**
IEOR Department
Columbia University
vg2277@columbia.edu

## Abstract

Affine policies (or control) are widely used as a solution approach in dynamic optimization where computing an optimal adjustable solution is usually intractable. While the worst case performance of affine policies can be significantly bad, the empirical performance is observed to be near-optimal for a large class of problem instances. For instance, in the two-stage dynamic robust optimization problem with linear covering constraints and uncertain right hand side, the worst-case approximation bound for affine policies is $O(\sqrt{m})$ that is also tight (see Bertsimas and Goyal [8]), whereas observed empirical performance is near-optimal. In this paper, we aim to address this stark-contrast between the worst-case and the empirical performance of affine policies. In particular, we show that affine policies give a good approximation for the two-stage adjustable robust optimization problem with high probability on random instances where the constraint coefficients are generated i.i.d. from a large class of distributions; thereby, providing a theoretical justification of the observed empirical performance. On the other hand, we also present a distribution such that the performance bound for affine policies on instances generated according to that distribution is $\Omega(\sqrt{m})$ with high probability; however, the constraint coefficients are not i.i.d.. This demonstrates that the empirical performance of affine policies can depend on the generative model for instances.

## 1 Introduction

In most real word problems, parameters are uncertain at the optimization phase and decisions need to be made in the face of uncertainty. Stochastic and robust optimization are two widely used paradigms to handle uncertainty. In the stochastic optimization approach, uncertainty is modeled as a probability distribution and the goal is to optimize an expected objective [13]. We refer the reader to Kall and Wallace [19], Prekopa [20], Shapiro [21], Shapiro et al. [22] for a detailed discussion on stochastic optimization. On the other hand, in the robust optimization approach, we consider an adversarial model of uncertainty using an uncertainty set and the goal is to optimize over the worst-case realization from the uncertainty set. This approach was first introduced by Soyster [23] and has been extensively studied in recent past. We refer the reader to Ben-Tal and Nemirovski [3, 4, 5], El Ghaoui and Lebret [14], Bertsimas and Sim [10, 11], Goldfarb and Iyengar [17], Bertsimas et al. [6] and Ben-Tal et al. [1] for a detailed discussion of robust optimization. However, in both these paradigms, computing an optimal dynamic solution is intractable in general due to the "curse of dimensionality".

This intractability of computing the optimal adjustable solution necessitates considering approximate solution policies such as static and affine policies where the decision in any period $t$ is restricted to a particular function of the sample path until period $t$. Both static and affine policies have been

studied extensively in the literature and can be computed efficiently for a large class of problems. While the worst-case performance of such approximate policies can be significantly bad as compared to the optimal dynamic solution, the empirical performance, especially of affine policies, has been observed to be near-optimal in a broad range of computational experiments. Our goal in this paper is to address this stark contrast between the worst-case performance bounds and near-optimal empirical performance of affine policies.

In particular, we consider the following two-stage adjustable robust linear optimization problems with uncertain demand requirements:

$$z_{\mathsf{AR}}\left(\boldsymbol{c},\boldsymbol{d},\boldsymbol{A},\boldsymbol{B},\mathcal{U}\right) = \min_{\boldsymbol{x}} \boldsymbol{c}^T\boldsymbol{x} + \max_{\boldsymbol{h}\in\mathcal{U}}\min_{\boldsymbol{y}(\boldsymbol{h})}\boldsymbol{d}^T\boldsymbol{y}(\boldsymbol{h})$$
$$\boldsymbol{A}\boldsymbol{x} + \boldsymbol{B}\boldsymbol{y}(\boldsymbol{h}) \ \geq \ \boldsymbol{h} \quad \forall\boldsymbol{h}\in\mathcal{U} \tag{1}$$
$$\boldsymbol{x}\in\mathbb{R}^n_+, \ \boldsymbol{y}(\boldsymbol{h})\in\mathbb{R}^n_+ \ \forall\boldsymbol{h}\in\mathcal{U}$$

where $\boldsymbol{A}\in\mathbb{R}^{m\times n}_+, \boldsymbol{c}\in\mathbb{R}^n_+, \boldsymbol{d}\in\mathbb{R}^n_+, \boldsymbol{B}\in\mathbb{R}^{m\times n}_+$. The right-hand-side $\boldsymbol{h}$ belongs to a compact convex uncertainty set $\mathcal{U}\subseteq\mathbb{R}^m_+$. The goal in this problem is to select the first-stage decision $\boldsymbol{x}$, and the second-stage recourse decision, $\boldsymbol{y}(\boldsymbol{h})$, as a function of the uncertain right hand side realization, $\boldsymbol{h}$ such that the worst-case cost over all realizations of $\boldsymbol{h}\in\mathcal{U}$ is minimized. We assume without loss of generality that $\boldsymbol{c}=\boldsymbol{e}$ and $\boldsymbol{d}=\bar{d}\cdot\boldsymbol{e}$ (by appropriately scaling $\boldsymbol{A}$ and $\boldsymbol{B}$). Here, $\bar{d}$ can interpreted as the inflation factor for costs in the second-stage.

This model captures many important applications including set cover, facility location, network design, inventory management, resource planning and capacity planning under uncertain demand. Here the right hand side, $\boldsymbol{h}$ models the uncertain demand and the covering constraints capture the requirement of satisfying the uncertain demand. However, the adjustable robust optimization problem (1) is intractable in general. In fact, Feige et al. [16] show that $\Pi_{\mathsf{AR}}(\mathcal{U})$ (1) is hard to approximate within any factor that is better than $\Omega(\log n)$.

Both static and affine policy approximations have been studied in the literature for (1). In a static solution, we compute a single optimal solution $(\boldsymbol{x},\boldsymbol{y})$ that is feasible for all realizations of the uncertain right hand side. Bertsimas et al. [9] relate the performance of static solution to the symmetry of the uncertainty set and show that it provides a good approximation to the adjustable problem if the uncertainty is close to being centrally symmetric. However, the performance of static solutions can be arbitrarily large for a general convex uncertainty set with the worst case performance being $\Omega(m)$. El Housni and Goyal [15] consider piecewise static policies for two-stage adjustable robust problem with uncertain constraint coefficients. These are a generalization of static policies where we divide the uncertainty set into several pieces and specify a static solution for each piece. However, they show that, in general, there is no piecewise static policy with a polynomial number of pieces that has a significantly better performance than an optimal static policy.

An affine policy restricts the second-stage decisions, $\boldsymbol{y}(\boldsymbol{h})$ to being an affine function of the uncertain right-hand-side $\boldsymbol{h}$, i.e., $\boldsymbol{y}(\boldsymbol{h})=\boldsymbol{P}\boldsymbol{h}+\boldsymbol{q}$ for some $\boldsymbol{P}\in\mathbb{R}^{n\times m}$ and $\boldsymbol{q}\in\mathbb{R}^m$ are decision variables. Affine policies in this context were introduced in Ben-Tal et al. [2] and can be formulated as:

$$z_{\mathsf{Aff}}\left(\boldsymbol{c},\boldsymbol{d},\boldsymbol{A},\boldsymbol{B},\mathcal{U}\right) = \min_{\boldsymbol{x},\boldsymbol{P},\boldsymbol{q}} \boldsymbol{c}^T\boldsymbol{x} + \max_{\boldsymbol{h}\in\mathcal{U}}\boldsymbol{d}^T\left(\boldsymbol{P}\boldsymbol{h}+\boldsymbol{q}\right)$$
$$\boldsymbol{A}\boldsymbol{x} + \boldsymbol{B}\left(\boldsymbol{P}\boldsymbol{h}+\boldsymbol{q}\right) \ \geq \ \boldsymbol{h} \quad \forall\boldsymbol{h}\in\mathcal{U} \tag{2}$$
$$\boldsymbol{P}\boldsymbol{h}+\boldsymbol{q} \ \geq \ \boldsymbol{0} \quad \forall\boldsymbol{h}\in\mathcal{U}$$
$$\boldsymbol{x}\in\mathbb{R}^n_+$$

An optimal affine policy can be computed efficiently for a large class of problems. Bertsimas and Goyal [8] show that affine policies give a $O(\sqrt{m})$-approximation to the optimal dynamic solution for (1). Furthermore, they show that the approximation bound $O(\sqrt{m})$ is tight. However, the observed empirical performance for affine policies is near-optimal for a large set of synthetic instances of (1).

## 1.1  Our Contributions

Our goal in this paper is to address this stark contrast by providing a theoretical analysis of the performance of affine policies on synthetic instances of the problem generated from a probabilistic model. In particular, we consider random instances of the two-stage adjustable problem (1) where the entries of the constraint matrix $\boldsymbol{B}$ are random from a given distribution and analyze the performance of affine policies for a large class of distributions. Our main contributions are summarized below.

**Independent and Identically distributed Constraint Coefficients**. We consider random instances of the two-stage adjustable problem where the entries of $\boldsymbol{B}$ are generated i.i.d. according to a given distribution and show that an affine policy gives a good approximation for a large class of distributions including distributions with bounded support and unbounded distributions with Gaussian and sub-gaussian tails.

In particular, for distributions with bounded support in $[0, b]$ and expectation $\mu$, we show that for sufficiently large values of $m$ and $n$, affine policy gives a $b/\mu$-approximation to the adjustable problem (1). More specifically, with probability at least $(1 - 1/m)$, we have that

$$z_{\mathsf{Aff}}(\boldsymbol{c}, \boldsymbol{d}, \boldsymbol{A}, \boldsymbol{B}, \mathcal{U}) \leq \frac{b}{\mu(1 - \epsilon)} \cdot z_{\mathsf{AR}}(\boldsymbol{c}, \boldsymbol{d}, \boldsymbol{A}, \boldsymbol{B}, \mathcal{U}),$$

where $\epsilon = b/\mu\sqrt{\log m/n}$ (Theorem 2.1). Therefore, if the distribution is *symmetric*, affine policy gives a 2-approximation for the adjustable problem (1). For instance, for the case of uniform or Bernoulli distribution with parameter $p = 1/2$, affine gives a nearly 2-approximation for (1).

While the above bound leads to a good approximation for many distributions, the ratio $\frac{b}{\mu}$ can be significantly large in general; for instance, for distributions where extreme values of the support are extremely rare and significantly far from the mean. In such instances, the bound $b/\mu$ can be quite loose. We can tighten the analysis by using the concentration properties of distributions and can extend the analysis even for the case of unbounded support. More specifically, we show that if $B_{ij}$ are i.i.d. according to an unbounded distribution with a sub-gaussian tail, then for sufficiently large values of $m$ and $n$, with probability at least $(1 - 1/m)$,

$$z_{\mathsf{Aff}}(\boldsymbol{c}, \boldsymbol{d}, \boldsymbol{A}, \boldsymbol{B}, \mathcal{U}) \leq O(\sqrt{\log mn}) \cdot z_{\mathsf{AR}}(\boldsymbol{c}, \boldsymbol{d}, \boldsymbol{A}, \boldsymbol{B}, \mathcal{U}).$$

We prove the case of *folded normal* distribution in Theorem 2.6. Here we assume that the parameters of the distributions are constants independent of the problem dimension and we would like to emphasis that the i.i.d. assumption on the entries of $\boldsymbol{B}$ is for the scaled problem where $\boldsymbol{c} = \boldsymbol{e}$ and $\boldsymbol{d} = \tilde{d}\boldsymbol{e}$.

We would like to note that the above performance bounds are in stark contrast with the worst case performance bound $O(\sqrt{m})$ for affine policies which is tight. For the random instances where $B_{ij}$ are i.i.d. according to above distributions, the performance is significantly better. Therefore, our results provide a theoretical justification of the good empirical performance of affine policies and close the gap between worst case bound of $O(\sqrt{m})$ and observed empirical performance. Furthermore, surprisingly these performance bounds are independent of the structure of the uncertainty set, $\mathcal{U}$ unlike in previous work where the performance bounds depend on the geometric properties of $\mathcal{U}$. Our analysis is based on a *dual-reformulation* of (1) introduced in [7] where (1) is reformulated as an alternate two-stage adjustable optimization and the uncertainty set in the alternate formulation depends on the constraint matrix $\boldsymbol{B}$. Using the probabilistic structure of $\boldsymbol{B}$, we show that the alternate *dual* uncertainty set is close to a simplex for which affine policies are optimal.

We would also like to note that our performance bounds are not necessarily tight and the actual performance on particular instances can be even better. We test the empirical performance of affine policies for random instances generated according to uniform and folded normal distributions and observe that affine policies are nearly optimal with a worst optimality gap of $4\%$ (i.e. approximation ratio of $1.04$) on our test instances as compared to the optimal adjustable solution that is computed using a Mixed Integer Program (MIP).

**Worst-case distribution for Affine policies.** While for a large class of commonly used distributions, affine policies give a good approximation with high probability for random i.i.d. instances according to the given distribution, we present a distribution where the performance of affine policies is $\Omega(\sqrt{m})$ with high probability for instances generated from this distribution. Note that this matches the worst-case deterministic bound for affine policies. We would like to remark that in the worst-case distribution, the coefficients $B_{ij}$ are not identically distributed. Our analysis suggests that to obtain bad instances for affine policies, we need to generate instances using a structured distribution where the structure of the distribution might depend on the problem structure.

## 2 Random instances with i.i.d. coefficients

In this section, we theoretically characterize the performance of affine policies for random instances of (1) for a large class of generative distributions including both bounded and unbounded support

distributions. In particular, we consider the two-stage problem where constraint coefficients $\boldsymbol{A}$ and $\boldsymbol{B}$ are i.i.d. according to a given distribution. We consider a polyhedral uncertainty set $\mathcal{U}$ given as

$$\mathcal{U} = \{\boldsymbol{h} \in \mathbb{R}^m_+ \mid \boldsymbol{R}\boldsymbol{h} \leq \boldsymbol{r}\} \tag{3}$$

where $\boldsymbol{R} \in \mathbb{R}^{L \times m}_+$ and $\boldsymbol{r} \in \mathbb{R}^L_+$. This is a fairly general class of uncertainty sets that includes many commonly used sets such as hypercube and *budget uncertainty* sets.

Our analysis of the performance of affine policies does not depend on the structure of first stage constraint matrix $\boldsymbol{A}$ or cost $\boldsymbol{c}$. The second-stage cost, as already mentioned, is wlog of the form $\boldsymbol{d} = \bar{d}\boldsymbol{e}$. Therefore, we restrict our attention only to the distribution of coefficients of the second stage matrix $\boldsymbol{B}$. We will use the notation $\tilde{\boldsymbol{B}}$ to emphasis that $\boldsymbol{B}$ is random. For simplicity, we refer to $z_{\mathsf{AR}}(\boldsymbol{c}, \boldsymbol{d}, \boldsymbol{A}, \boldsymbol{B}, \mathcal{U})$ as $z_{\mathsf{AR}}(\boldsymbol{B})$ and to $z_{\mathsf{Aff}}(\boldsymbol{c}, \boldsymbol{d}, \boldsymbol{A}, \boldsymbol{B}, \mathcal{U})$ as $z_{\mathsf{Aff}}(\boldsymbol{B})$.

## 2.1 Distributions with bounded support

We first consider the case when $\tilde{B}_{ij}$ are i.i.d. according to a bounded distribution with support in $[0, b]$ for some constant $b$ independent of the dimension of the problem. We show a performance bound of affine policies as compared to the optimal dynamic solution. The bound depends only on the distribution of $\tilde{\boldsymbol{B}}$ and holds for any polyhedral uncertainty set $\mathcal{U}$. In particular, we have the following theorem.

**Theorem 2.1.** *Consider the two-stage adjustable problem* (1) *where $\tilde{B}_{ij}$ are i.i.d. according to a bounded distribution with support in $[0, b]$ and $\mathbb{E}[\tilde{B}_{ij}] = \mu \; \forall i \in [m] \; \forall j \in [n]$. For $n$ and $m$ sufficiently large, we have with probability at least $1 - \frac{1}{m}$,*

$$z_{\mathsf{Aff}}(\tilde{\boldsymbol{B}}) \leq \frac{b}{\mu(1-\epsilon)} \cdot z_{\mathsf{AR}}(\tilde{\boldsymbol{B}})$$

*where $\epsilon = \frac{b}{\mu}\sqrt{\frac{\log m}{n}}$.*

The above theorem shows that for sufficiently large values of $m$ and $n$, the performance of affine policies is at most $b/\mu$ times the performance of an optimal adjustable solution. Moreover, we know that $z_{\mathsf{AR}}(\tilde{\boldsymbol{B}}) \leq z_{\mathsf{Aff}}(\tilde{\boldsymbol{B}})$ for any $\boldsymbol{B}$ since the adjustable problem is a relaxation of the affine problem. This shows that affine policies give a good approximation (and significantly better than the worst-case bound of $O(\sqrt{m})$) for many important distributions. We present some examples below.

**Example 1. [Uniform distribution]** Suppose for all $i \in [m]$ and $j \in [n]$ $\tilde{B}_{ij}$ are i.i.d. uniform in $[0, 1]$. Then $\mu = 1/2$ and from Theorem 2.1 we have with probability at least $1 - 1/m$,

$$z_{\mathsf{Aff}}(\tilde{\boldsymbol{B}}) \leq \frac{2}{1-\epsilon} \cdot z_{\mathsf{AR}}(\tilde{\boldsymbol{B}})$$

where $\epsilon = 2\sqrt{\log m/n}$. Therefore, for sufficiently large values of $n$ and $m$ affine policy gives a 2-approximation to the adjustable problem in this case. Note that the approximation bound of 2 is a conservative bound and the empirical performance is significantly better. We demonstrate this in our numerical experiments.

**Example 2. [Bernoulli distribution]** Suppose for all $i \in [m]$ and $j \in [n]$, $\tilde{B}_{ij}$ are i.i.d. according to a Bernoulli distribution of parameter $p$. Then $\mu = p$, $b = 1$ and from Theorem 2.1 we have with probability at least $1 - \frac{1}{m}$,

$$z_{\mathsf{Aff}}(\tilde{\boldsymbol{B}}) \leq \frac{1}{p(1-\epsilon)} \cdot z_{\mathsf{AR}}(\tilde{\boldsymbol{B}})$$

where $\epsilon = \frac{1}{p}\sqrt{\frac{\log m}{n}}$. Therefore for constant $p$, affine policy gives a constant approximation to the adjustable problem (for example 2-approximation for $p = 1/2$).

Note that these performance bounds are in stark contrast with the worst case performance bound $O(\sqrt{m})$ for affine policies which is tight. For these random instances, the performance is significantly better. We would like to note that the above distributions are very commonly used to generate instances for testing the performance of affine policies and exhibit good empirical performance.

Here, we give a theoretical justification of the good empirical performance of affine policies on such instances, thereby closing the gap between worst case bound of $O(\sqrt{m})$ and observed empirical performance. We discuss the intuition and the proof of Theorem 2.1 in the following subsections.

### 2.1.1 Preliminaries

In order to prove Theorem 2.1, we need to introduce certain preliminary results. We first introduce the following formulation for the adjustable problem (1) based on ideas in Bertsimas and de Ruiter [7].

$$z_{\text{d}-\text{AR}}(\boldsymbol{B}) = \min_{\boldsymbol{x}} \boldsymbol{c}^T \boldsymbol{x} + \max_{\boldsymbol{w} \in \mathcal{W}} \min_{\boldsymbol{\lambda}(\boldsymbol{w})} -(\boldsymbol{A}\boldsymbol{x})^T \boldsymbol{w} + \boldsymbol{r}^T \boldsymbol{\lambda}(\boldsymbol{w})$$

$$\boldsymbol{R}^T \boldsymbol{\lambda}(\boldsymbol{w}) \geq \boldsymbol{w} \quad \forall \boldsymbol{w} \in \mathcal{W} \tag{4}$$

$$\boldsymbol{x} \in \mathbb{R}^n_+, \ \boldsymbol{\lambda}(\boldsymbol{w}) \in \mathbb{R}^L_+, \ \forall \boldsymbol{w} \in \mathcal{W}$$

where the set $\mathcal{W}$ is defined as

$$\mathcal{W} = \{\boldsymbol{w} \in \mathbb{R}^m_+ \mid \boldsymbol{B}^T \boldsymbol{w} \leq \boldsymbol{d}\}. \tag{5}$$

We show that the above problem is an equivalent formulation of (1).

**Lemma 2.2.** *Let $z_{\text{AR}}(\boldsymbol{B})$ be as defined in (1) and $z_{\text{d}-\text{AR}}(\boldsymbol{B})$ as defined in (4). Then, $z_{\text{AR}}(\boldsymbol{B}) = z_{\text{d}-\text{AR}}(\boldsymbol{B})$.*

The proof follows from [7]. For completeness, we present it in Appendix A. Reformulation (4) can be interpreted as a new two-stage adjustable problem over *dualized* uncertainty set $\mathcal{W}$ and decision $\boldsymbol{\lambda}(\boldsymbol{w})$. Following [7], we refer to (4) as the *dualized* formulation and to (1) as the *primal* formulation. Bertsimas and de Ruiter [7] show that even the affine approximations of (1) and (4) (where recourse decisions are restricted to be affine functions of respective uncertainties) are equivalent. In particular, we have the following Lemma which is a restatement of Theorem 2 in [7].

**Lemma 2.3.** **(Theorem 2 in Bertsimas and de Ruiter [7])** *Let $z_{\text{d}-\text{Aff}}(\boldsymbol{B})$ be the objective value when $\boldsymbol{\lambda}(\boldsymbol{w})$ is restricted to be affine function of $\boldsymbol{w}$ and $z_{\text{Aff}}(\boldsymbol{B})$ as defined in (2). Then $z_{\text{d}-\text{Aff}}(\boldsymbol{B}) = z_{\text{Aff}}(\boldsymbol{B})$.*

Bertsimas and Goyal [8] show that affine policy is optimal for the adjustable problem (1) when the uncertainty set $\mathcal{U}$ is a simplex. In fact, optimality of affine policies for simplex uncertainty sets holds for more general formulation than considered in [8]. In particular, we have the following lemma

**Lemma 2.4.** *Suppose the set $\mathcal{W}$ is a simplex, i.e. a convex combination of $m+1$ affinely independent points, then affine policy is optimal for the adjustable problem (4), i.e. $z_{\text{d}-\text{Aff}}(\boldsymbol{B}) = z_{\text{d}-\text{AR}}(\boldsymbol{B})$.*

The proof proceeds along similar lines as in [8]. For completeness, we provide it in Appendix A. In fact, if the uncertainty set is not simplex but can be approximated by a simplex within a small scaling factor, affine policies can still be shown to be a good approximation, in particular we have the following lemma.

**Lemma 2.5.** *Denote $\mathcal{W}$ the dualized uncertainty set as defined in (5) and suppose there exists a simplex $\mathcal{S}$ and $\kappa \geq 1$ such that $\mathcal{S} \subseteq \mathcal{W} \subseteq \kappa \cdot \mathcal{S}$. Therefore, $z_{\text{d}-\text{AR}}(\boldsymbol{B}) \leq z_{\text{d}-\text{Aff}}(\boldsymbol{B}) \leq \kappa \cdot z_{\text{d}-\text{AR}}(\boldsymbol{B})$. Furthermore, $z_{\text{AR}}(\boldsymbol{B}) \leq z_{\text{Aff}}(\boldsymbol{B}) \leq \kappa \cdot z_{\text{AR}}(\boldsymbol{B})$.*

The proof of Lemma 2.5 is presented in Appendix A.

### 2.1.2 Proof of Theorem 2.1

We consider instances of problem (1) where $\tilde{B}_{ij}$ are i.i.d. according to a bounded distribution with support in $[0, b]$ and $\mathbb{E}[\tilde{B}_{ij}] = \mu$ for all $i \in [m], j \in [n]$. Denote the dualized uncertainty set $\tilde{\mathcal{W}} = \{\boldsymbol{w} \in \mathbb{R}^m_+ \mid \tilde{\boldsymbol{B}}^T \boldsymbol{w} \leq \bar{d} \cdot \boldsymbol{e}\}$. Our performance bound is based on showing that $\tilde{\mathcal{W}}$ can be sandwiched between two simplices with a small scaling factor. In particular, consider the following simplex,

$$\mathcal{S} = \left\{ \boldsymbol{w} \in \mathbb{R}^m_+ \ \middle| \ \sum_{i=1}^m w_i \leq \frac{\bar{d}}{b} \right\}. \tag{6}$$

we will show that $\mathcal{S} \subseteq \tilde{\mathcal{W}} \subseteq \frac{b}{\mu(1-\epsilon)} \cdot \mathcal{S}$ with probability at least $1 - \frac{1}{m}$ where $\epsilon = \frac{b}{\mu} \sqrt{\frac{\log m}{n}}$.

First, we show that $\mathcal{S} \subseteq \tilde{\mathcal{W}}$. Consider any $\boldsymbol{w} \in \mathcal{S}$. For any any $i = 1, \ldots, n$

$$\sum_{j=1}^{m} \tilde{B}_{ji} w_j \leq b \sum_{j=1}^{m} w_j \leq \bar{d}$$

The first inequality holds because all components of $\tilde{\boldsymbol{B}}$ are upper bounded by $b$ and the second one follows from $\boldsymbol{w} \in \mathcal{S}$. Hence, we have $\tilde{\boldsymbol{B}}^T \boldsymbol{w} \leq \bar{d}\boldsymbol{e}$ and consequently $\mathcal{S} \subseteq \tilde{\mathcal{W}}$.

Now, we show that the other inclusion holds with high probability. Consider any $\boldsymbol{w} \in \tilde{\mathcal{W}}$. We have $\tilde{\boldsymbol{B}}^T \boldsymbol{w} \leq \bar{d} \cdot \boldsymbol{e}$. Summing up all the inequalities and dividing by $n$, we get

$$\sum_{j=1}^{m} \left( \frac{\sum_{i=1}^{n} \tilde{B}_{ji}}{n} \right) \cdot w_j \leq \bar{d}. \tag{7}$$

Using Hoeffding's inequality [18] (see Appendix B) with $\tau = b\sqrt{\frac{\log m}{n}}$, we have

$$\mathbb{P} \left( \frac{\sum_{i=1}^{n} \tilde{B}_{ji}}{n} - \mu \geq -\tau \right) \geq 1 - \exp\left( \frac{-2n\tau^2}{b^2} \right) = 1 - \frac{1}{m^2}$$

and a union bound over $j = 1, \ldots, m$ gives us

$$\mathbb{P} \left( \frac{\sum_{i=1}^{n} \tilde{B}_{ji}}{n} \geq \mu - \tau \ \ \forall j = 1, \ldots, m \right) \geq \left( 1 - \frac{1}{m^2} \right)^m \geq 1 - \frac{1}{m}.$$

where the last inequality follows from Bernoulli's inequality. Therefore, with probability at least $1 - \frac{1}{m}$, we have

$$\sum_{j=1}^{m} w_j \leq \sum_{j=1}^{m} \frac{1}{\mu - \tau} \left( \frac{\sum_{i=1}^{n} \tilde{B}_{ji}}{n} \right) \cdot w_j \leq \frac{\bar{d}}{(\mu - \tau)} = \frac{b}{\mu(1 - \epsilon)} \cdot \frac{\bar{d}}{b}$$

where the second inequality follows from (7). Note that for $m$ sufficiently large , we have $\mu - \tau > 0$. Then, $\boldsymbol{w} \in \frac{b}{\mu(1-\epsilon)} \cdot \mathcal{S}$ for any $\boldsymbol{w} \in \tilde{\mathcal{W}}$ and consequently $\mathcal{S} \subseteq \tilde{\mathcal{W}} \subseteq \frac{b}{\mu(1-\epsilon)} \cdot \mathcal{S}$ with probability at least $1 - 1/m$. Finally, we apply the result of Lemma 2.5 to conclude. $\qquad\square$

## 2.2 Unbounded distributions

While the approximation bound in Theorem 2.1 leads to a good approximation for many distributions, the ratio $b/\mu$ can be significantly large in general. We can tighten the analysis by using the concentration properties of distributions and can extend the analysis even for the case of distributions with unbounded support and sub-gaussian tails. In this section, we consider the special case where $\tilde{B}_{ij}$ are i.i.d. according to absolute value of a standard Gaussian, also called the *folded normal* distribution, and show a logarithmic approximation bound for affine policies. In particular, we have the following theorem.

**Theorem 2.6.** *Consider the two-stage adjustable problem* (1) *where $\forall i \in [n], j \in [m]$, $\tilde{B}_{ij} = |\tilde{G}_{ij}|$ and $\tilde{G}_{ij}$ are i.i.d. according to a standard Gaussian distribution. For $n$ and $m$ sufficiently large, we have with probability at least $1 - \frac{1}{m}$,*

$$z_{\mathsf{Aff}}(\tilde{\boldsymbol{B}}) \leq \kappa \cdot z_{\mathsf{AR}}(\tilde{\boldsymbol{B}})$$

*where $\kappa = O\left(\sqrt{\log m + \log n}\right)$.*

The proof of Theorem 2.6 is presented in Appendix C. We can extend the analysis and show a similar bound for the class of distributions with sub-gaussian tails. The bound of $O\left(\sqrt{\log m + \log n}\right)$ depends on the dimension of the problem unlike the case of uniform bounded distribution. But, it is significantly better than the worst-case of $O(\sqrt{m})$ [8] for general instances. Furthermore, this bound holds for all uncertainty sets with high probability. We would like to note though that the bounds are not necessarily tight. In fact, in our numerical experiments where the uncertainty set is a *budget of uncertainty*, we observe that affine policies are near optimal.

## 3 Family of worst-case distribution: perturbation of i.i.d. coefficients

For any $m$ sufficiently large, the authors in [8] present an instance where affine policy is $\Omega(m^{\frac{1}{2}-\delta})$ away from the optimal adjustable solution. The parameters of the instance in [8] were carefully chosen to achieve the gap $\Omega(m^{\frac{1}{2}-\delta})$. In this section, we show that the family of worst-case instances is not measure zero set. In fact, we exhibit a distribution and an uncertainty set such that a random instance from that distribution achieves a worst-case bound of $\Omega(\sqrt{m})$ with high probability. The coefficients $\tilde{B}_{ij}$ in our bad family of instances are independent but not identically distributed. The instance can be given as follows.

$$n = m, \quad \boldsymbol{A} = \boldsymbol{0}, \quad \boldsymbol{c} = \boldsymbol{0}, \quad \boldsymbol{d} = \boldsymbol{e}$$

$$\mathcal{U} = \mathsf{conv}\left(\boldsymbol{0}, \boldsymbol{e}_1, \ldots, \boldsymbol{e}_m, \boldsymbol{\nu}_1, \ldots, \boldsymbol{\nu}_m\right) \quad \text{where } \boldsymbol{\nu}_i = \frac{1}{\sqrt{m}}(\boldsymbol{e} - \boldsymbol{e}_i)\ \forall i \in [m]. \tag{8}$$

$$\tilde{B}_{ij} = \begin{cases} 1 & \text{if } i = j \\ \frac{1}{\sqrt{m}} \cdot \tilde{u}_{ij} & \text{if } i \neq j \end{cases} \quad \text{where for all } i \neq j, \tilde{u}_{ij} \text{ are i.i.d. uniform}[0,1].$$

**Theorem 3.1.** *For the instance defined in* (8)*, we have with probability at least* $1 - 1/m$,

$$z_{\mathsf{Aff}}(\tilde{\boldsymbol{B}}) = \Omega(\sqrt{m}) \cdot z_{\mathsf{AR}}(\tilde{\boldsymbol{B}}).$$

We present the proof of Theorem 3.1 in Appendix D. As a byproduct, we also tighten the lower bound on the performance of affine policy to $\Omega(\sqrt{m})$ improving from the lower bound of $\Omega(m^{\frac{1}{2}-\delta})$ in [8]. We would like to note that both uncertainty set and distribution of coefficients in our instance (8) are carefully chosen to achieve the worst-case gap. Our analysis suggests that to obtain bad instances for affine policies, we need to generate instances using a structured distribution as above and it may not be easy to obtain bad instances in a completely random setting.

## 4 Performance of affine policy: Empirical study

In this section, we present a computational study to test the empirical performance of affine policy for the two-stage adjustable problem (1) on random instances.

**Experimental setup.** We consider two classes of distributions for generating random instances: $i)$ Coefficients of $\tilde{\boldsymbol{B}}$ are i.i.d. uniform $[0,1]$, and $ii)$ Coefficients of $\tilde{\boldsymbol{B}}$ are absolute value of i.i.d. standard Gaussian. We consider the following *budget of uncertainty* set.

$$\mathcal{U} = \left\{ \boldsymbol{h} \in [0,1]^m \ \middle| \ \sum_{i=1}^m h_i \leq \sqrt{m} \right\}. \tag{9}$$

Note that the set (9) is widely used in both theory and practice and arises naturally as a consequence of concentration of sum of independent uncertain demand requirements. We would like to also note that the adjustable problem over this budget of uncertainty, $\mathcal{U}$ is hard to approximate within a factor better than $O(\log n)$ [16]. We consider $n = m, \boldsymbol{d} = \boldsymbol{e}$. Also, we consider $\boldsymbol{c} = \boldsymbol{0}, \boldsymbol{A} = \boldsymbol{0}$. We restrict to this case in order to compute the optimal adjustable solution in a reasonable time by solving a single Mixed Integer Program (MIP). For the general problem, computing the optimal adjustable solution requires solving a sequence of MIPs each one of which is significantly challenging to solve. We would like to note though that our analysis does not depend on the first stage cost $\boldsymbol{c}$ and matrix $\boldsymbol{A}$ and affine policy can be computed efficiently even without this assumption. We consider values of $m$ from 10 to 50 and consider 20 instances for each value of $m$. We report the ratio $r = z_{\mathsf{Aff}}(\tilde{\boldsymbol{B}})/z_{\mathsf{AR}}(\tilde{\boldsymbol{B}})$ in Table 1. In particular, for each value of $m$, we report the average ratio $r_{\mathsf{avg}}$, the maximum ratio $r_{\mathsf{max}}$, the running time of adjustable policy $T_{\mathsf{AR}}(s)$ and the running time of affine policy $T_{\mathsf{Aff}}(s)$. We first give a compact LP formulation for the affine problem (2) and a compact MIP formulation for the separation of the adjustable problem(1).

**LP formulations for the affine policies.** The affine problem (2) can be reformulated as follows

$$z_{\mathsf{Aff}}(\boldsymbol{B}) = \min \left\{ \boldsymbol{c}^T \boldsymbol{x} + z \ \middle| \ \begin{array}{ll} z \geq \boldsymbol{d}^T \left(\boldsymbol{Ph} + \boldsymbol{q}\right) & \forall \boldsymbol{h} \in \mathcal{U} \\ \boldsymbol{Ax} + \boldsymbol{B}\left(\boldsymbol{Ph} + \boldsymbol{q}\right) \geq \boldsymbol{h} & \forall \boldsymbol{h} \in \mathcal{U} \\ \boldsymbol{Ph} + \boldsymbol{q} \geq \boldsymbol{0} & \forall \boldsymbol{h} \in \mathcal{U} \\ \boldsymbol{x} \in \mathbb{R}^n_+ \end{array} \right\}.$$

Note that this formulation has infinitely many constraints but we can write a compact LP formulation using standard techniques from duality. For example, the first constraint is equivalent to $z - \boldsymbol{d}^T \boldsymbol{q} \geq$ max $\{\boldsymbol{d}^T \boldsymbol{Ph} \mid \boldsymbol{Rh} \leq \boldsymbol{r}, \ \boldsymbol{h} \geq \boldsymbol{0}\}$. By taking the dual of the maximization problem, the constraint becomes $z - \boldsymbol{d}^T \boldsymbol{q} \geq$ min $\{\boldsymbol{r}^T \boldsymbol{v} \mid \boldsymbol{R}^T \boldsymbol{v} \geq \boldsymbol{P}^T \boldsymbol{d}, \ \boldsymbol{v} \geq \boldsymbol{0}\}$. We can then drop the min and introduce $\boldsymbol{v}$ as a variable, hence we obtain the following linear constraints $z - \boldsymbol{d}^T \boldsymbol{q} \geq \boldsymbol{r}^T \boldsymbol{v}$ , $\boldsymbol{R}^T \boldsymbol{v} \geq \boldsymbol{P}^T \boldsymbol{d}$ and $\boldsymbol{v} \geq \boldsymbol{0}$. We can apply the same techniques for the other constraints. The complete LP formulation and its proof of correctness is presented in Appendix E.

**Mixed Integer Program Formulation for the adjustable problem** (1). For the adjustable problem (1), we show that the separation problem (10) can be formulated as a mixed integer program. The separation problem can be formulated as follows: Given $\hat{\boldsymbol{x}}$ and $\hat{z}$ decide whether

$$\max \ \{(\boldsymbol{h} - \boldsymbol{A}\hat{\boldsymbol{x}})^T \boldsymbol{w} \mid \boldsymbol{w} \in \mathcal{W}, \boldsymbol{h} \in \mathcal{U}\} > \hat{z} \tag{10}$$

The correctness of formulation (10) follows from equation (11) in the proof of Lemma 2.2 in Appendix A. The constraints in (10) are linear but the objective function contains a bilinear term, $\boldsymbol{h}^T \boldsymbol{w}$. We linearize this using a standard *digitized reformulation*. In particular, we consider finite bit representations of continuous variables, $h_i$ nd $w_i$ to desired accuracy and introduce additional binary variables, $\alpha_{ik}, \beta_{ik}$ where $\alpha_{ik}$ and $\beta_{ik}$ represents the $k^{th}$ bits of $h_i$ and $w_i$ respectively. Now, for any $i \in [m]$, $h_i \cdot w_i$ can be expressed as a bilinear expression with products of binary variables, $\alpha_{ik} \cdot \beta_{ij}$ which can be linearized using additional variable $\gamma_{ijk}$ and standard linear inequalities: $\gamma_{ijk} \leq \beta_{ij}$, $\gamma_{ijk} \leq \alpha_{ik}$, $\gamma_{ijk} + 1 \geq \alpha_{ik} + \beta_{ij}$. The complete MIP formulation and the proof of correctness is presented in Appendix E.

For general $\boldsymbol{A} \neq 0$, we need to solve a sequence of MIPs to find the optimal adjustable solution. In order to compute the optimal adjustable solution in a reasonable time, we assume $\boldsymbol{A} = 0, \boldsymbol{c} = 0$ in our experimental setting so that we only need to solve one MIP.

**Results.** In our experiments, we observe that the empirical performance of affine policy is near-optimal. In particular, the performance is significantly better than the theoretical performance bounds implied in Theorem 2.1 and Theorem 2.6. For instance, Theorem 2.1 implies that affine policy is a 2-approximation with high probability for random instances from a uniform distribution. However, in our experiments, we observe that the optimality gap for affine policies is at most $4\%$ (i.e. approximation ratio of at most 1.04). The same observation holds for Gaussian distributions as well Theorem 2.6 gives an approximation bound of $O(\sqrt{\log(mn)})$. We would like to remark that we are not able to report the ratio $r$ for large values of $m$ because the adjustable problem is computationally very challenging and for $m \geq 40$, MIP does not solve within a time limit of 3 hours for most instances . On the other hand, affine policy scales very well and the average running time is few seconds even for large values of $m$. This demonstrates the power of affine policies that can be computed efficiently and give good approximations for a large class of instances.

| $m$ | $r_{\text{avg}}$ | $r_{\text{max}}$ | $T_{\text{AR}}(s)$ | $T_{\text{Aff}}(s)$ |
|---|---|---|---|---|
| 10 | 1.01 | 1.03 | 10.55 | 0.01 |
| 20 | 1.02 | 1.04 | 110.57 | 0.23 |
| 30 | 1.01 | 1.02 | 761.21 | 1.29 |
| 50 | ** | ** | ** | 14.92 |

(a) Uniform

| $m$ | $r_{\text{avg}}$ | $r_{\text{max}}$ | $T_{\text{AR}}(s)$ | $T_{\text{Aff}}(s)$ |
|---|---|---|---|---|
| 10 | 1.00 | 1.03 | 12.95 | 0.01 |
| 20 | 1.01 | 1.03 | 217.08 | 0.39 |
| 30 | 1.01 | 1.03 | 594.15 | 1.15 |
| 50 | ** | ** | ** | 13.87 |

(b) Folded Normal

Table 1: Comparison on the performance and computation time of affine policy and optimal adjustable policy for uniform and folded normal distributions. For 20 instances, we compute $z_{\text{Aff}}(\tilde{\boldsymbol{B}})/z_{\text{AR}}(\tilde{\boldsymbol{B}})$ and present the average and max ratios. Here, $T_{\text{AR}}(s)$ denotes the running time for the adjustable policy and $T_{\text{Aff}}(s)$ denotes the running time for affine policy in seconds. ** Denotes the cases when we set a time limit of 3 hours. These results are obtained using Gurobi 7.0.2 on a 16-core server with 2.93GHz processor and 56GB RAM.

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
