[Supplementary Material]

# A  Proofs of preliminaries

## Proof of Lemma 2.2

**Proof.** We have

$$
\begin{aligned}
z_{\mathsf{AR}}(\boldsymbol{B}) &= \min_{\boldsymbol{x} \geq \boldsymbol{0}} \ \boldsymbol{c}^T \boldsymbol{x} + \max_{\boldsymbol{h} \in \mathcal{U}} \ \min_{\substack{\boldsymbol{B}\boldsymbol{y} \, \geq \, \boldsymbol{h} - \boldsymbol{A}\boldsymbol{x} \\ \boldsymbol{y} \geq \boldsymbol{0}}} \ \boldsymbol{d}^T \boldsymbol{y} \\
&= \min_{\boldsymbol{x} \geq \boldsymbol{0}} \ \boldsymbol{c}^T \boldsymbol{x} + \max_{\boldsymbol{h} \in \mathcal{U}} \ \max_{\substack{\boldsymbol{B}^T \boldsymbol{w} \leq \boldsymbol{d} \\ \boldsymbol{w} \geq \boldsymbol{0}}} \ (\boldsymbol{h} - \boldsymbol{A}\boldsymbol{x})^T \boldsymbol{w} && (11) \\
&= \min_{\boldsymbol{x} \geq \boldsymbol{0}} \ \boldsymbol{c}^T \boldsymbol{x} + \max_{\boldsymbol{w} \in \mathcal{W}} \ -(\boldsymbol{A}\boldsymbol{x})^T \boldsymbol{w} + \max_{\substack{\boldsymbol{R}\boldsymbol{h} \leq \boldsymbol{r} \\ \boldsymbol{h} \geq \boldsymbol{0}}} \ \boldsymbol{h}^T \boldsymbol{w} \\
&= \min_{\boldsymbol{x} \geq \boldsymbol{0}} \ \boldsymbol{c}^T \boldsymbol{x} + \max_{\boldsymbol{w} \in \mathcal{W}} \ -(\boldsymbol{A}\boldsymbol{x})^T \boldsymbol{w} + \min_{\substack{\boldsymbol{R}^T \boldsymbol{\lambda} \geq \boldsymbol{w} \\ \boldsymbol{\lambda} \geq \boldsymbol{0}}} \ \boldsymbol{r}^T \boldsymbol{\lambda} \\
&= z_{\mathsf{d-AR}}(\boldsymbol{B}).
\end{aligned}
$$

where the second equality holds by taking the dual of the inner minimization problem, the third equality follows from switching the two max, and the fourth one by taking the dual of the second maximization problem. □

## Proof of Lemma 2.4

**Proof.** We restate the same proof in [8] in our setting. First, since the adjustable problem is a relaxation of the affine problem then $z_{\mathsf{d-Aff}}(\boldsymbol{B}) \leq z_{\mathsf{d-AR}}(\boldsymbol{B})$.

Now let's prove the other inequality. Consider $\mathcal{W} = \{\boldsymbol{w} \in \mathbb{R}_+^m \mid \boldsymbol{B}^T \boldsymbol{w} \leq \boldsymbol{d}\}$ which is a simplex. Note that $\boldsymbol{0}$ is always an extreme point of the simplex $\mathcal{W}$ and denote $\boldsymbol{w}^1, \boldsymbol{w}^2, \ldots, \boldsymbol{w}^m$ the remaining $m$ points. In particular, we have for any $\boldsymbol{w} \in \mathcal{W}$

$$
\boldsymbol{w} = \sum_{j=1}^m \alpha_j \boldsymbol{w}^j = \boldsymbol{Q}\boldsymbol{\alpha}
$$

where $\sum_{j=1}^m \alpha_j \leq 1$ and $\boldsymbol{Q} = \left[\boldsymbol{w}^1 | \boldsymbol{w}^2 | \ldots | \boldsymbol{w}^m\right]$. Note that $\boldsymbol{Q}$ is invertible since $\boldsymbol{w}^1, \boldsymbol{w}^2, \ldots, \boldsymbol{w}^m$ are linearly independent. Hence, $\boldsymbol{\alpha} = \boldsymbol{Q}^{-1}\boldsymbol{w}$. Denote $\boldsymbol{x}^*, \boldsymbol{\lambda}^*(\boldsymbol{w})$ for $\boldsymbol{w} \in \mathcal{W}$ the optimal adjustable solution of the adjustable problem (4). We define the following affine solution $\boldsymbol{x} = \boldsymbol{x}^*$ and for $\boldsymbol{w} \in \mathcal{W}$, $\boldsymbol{\lambda}(\boldsymbol{w}) = \boldsymbol{P}\boldsymbol{Q}^{-1}\boldsymbol{w}$ where $\boldsymbol{P} = \left[\boldsymbol{\lambda}^*(\boldsymbol{w}^1) | \boldsymbol{\lambda}^*(\boldsymbol{w}^2) | \ldots | \boldsymbol{\lambda}^*(\boldsymbol{w}^m)\right]$. In particular, we have

$$
\boldsymbol{\lambda}(\boldsymbol{w}) = \sum_{j=1}^m \alpha_j \boldsymbol{\lambda}^*(\boldsymbol{w}^j).
$$

Let us first check the feasbility of the solution

$$
\boldsymbol{R}^T \boldsymbol{\lambda}(\boldsymbol{w}) = \sum_{j=1}^m \alpha_j \boldsymbol{R}^T \boldsymbol{\lambda}^*(\boldsymbol{w}^j) \geq \sum_{j=1}^m \alpha_j \boldsymbol{w}^j = \boldsymbol{w}
$$

where the inequality follows from the feasibility of the adjustable solution. Therefore,

$$
\begin{aligned}
z_{\mathsf{d-Aff}}(\boldsymbol{B}) &\leq \boldsymbol{c}^T \boldsymbol{x} + \max_{\boldsymbol{w} \in \mathcal{W}} \ (-\boldsymbol{A}\boldsymbol{x})^T \boldsymbol{w} + \boldsymbol{r}^T \boldsymbol{\lambda}(\boldsymbol{w}) \\
&= \boldsymbol{c}^T \boldsymbol{x}^* + \max_{\boldsymbol{\alpha}} \ (-\boldsymbol{A}\boldsymbol{x}^*)^T \boldsymbol{w} + \sum_{j=1}^m \alpha_j \boldsymbol{r}^T \boldsymbol{\lambda}^*(\boldsymbol{w}^j) \\
&= \boldsymbol{c}^T \boldsymbol{x}^* + \max_{\boldsymbol{\alpha}} \ \sum_{j=1}^m \alpha_j \left( (-\boldsymbol{A}\boldsymbol{x}^*)^T \boldsymbol{w}^j + \boldsymbol{r}^T \boldsymbol{\lambda}^*(\boldsymbol{w}^j) \right) \\
&\leq \boldsymbol{c}^T \boldsymbol{x}^* + \max_{\boldsymbol{w} \in \mathcal{W}} \left( (-\boldsymbol{A}\boldsymbol{x}^*)^T \boldsymbol{w} + \boldsymbol{r}^T \boldsymbol{\lambda}^*(\boldsymbol{w}) \right) \max_{\boldsymbol{\alpha}} \sum_{j=1}^m \alpha_j = z_{\mathsf{d-AR}}(\boldsymbol{B})
\end{aligned}
$$

where the last inequality holds because $\sum_{j=1}^m \alpha_j \leq 1$. We conclude that $z_{\mathsf{d-Aff}}(\boldsymbol{B}) = z_{\mathsf{d-AR}}(\boldsymbol{B})$. □

**Proof of Lemma 2.5**

**Proof.** First the inequality $z_{\text{d-AR}}(\boldsymbol{B}) \leq z_{\text{d-Aff}}(\boldsymbol{B})$ is straightforward since the adjustable problem(1) is a relaxation of the affine problem (2). On the other hand, since $\mathcal{W} \subseteq \kappa \cdot \mathcal{S}$ then,

$$z_{\text{d-Aff}}(\boldsymbol{B}) \leq \kappa \cdot z_{\text{d-Aff}}(\boldsymbol{B}, \mathcal{S})$$

where we denote $z_{\text{d-Aff}}(\boldsymbol{B}, \mathcal{S})$ the dualized affine problem over $\mathcal{S}$ (it's the same problem as $z_{\text{d-Aff}}(\boldsymbol{B})$ where we only replace $\mathcal{W}$ by $\mathcal{S}$). Since $\mathcal{S}$ is a simplex, from Lemma 2.4, we have $z_{\text{d-Aff}}(\boldsymbol{B}, \mathcal{S}) = z_{\text{d-AR}}(\boldsymbol{B}, \mathcal{S})$. Moreover, $z_{\text{d-AR}}(\boldsymbol{B}, \mathcal{S}) \leq z_{\text{d-AR}}(\boldsymbol{B})$ because $\mathcal{S} \subseteq \mathcal{W}$. We conclude that

$$z_{\text{d-AR}}(\boldsymbol{B}) \leq z_{\text{d-Aff}}(\boldsymbol{B}) \leq \kappa \cdot z_{\text{d-AR}}(\boldsymbol{B}).$$

Furthermore, since $z_{\text{d-AR}}(\boldsymbol{B}) = z_{\text{AR}}(\boldsymbol{B})$ from Lemma 2.2 and $z_{\text{d-Aff}}(\boldsymbol{B}) = z_{\text{Aff}}(\boldsymbol{B})$ from Lemma 2.3, then

$$z_{\text{AR}}(\boldsymbol{B}) \leq z_{\text{Aff}}(\boldsymbol{B}) \leq \kappa \cdot z_{\text{AR}}(\boldsymbol{B}).$$

$\square$

# B Hoeffding's inequality

**Hoeffding's inequality**[18]. Let $Z_1, \ldots, Z_n$ be independent bounded random variables with $Z_i \in [a, b]$ for all $i \in [n]$ and denote $Z = \frac{1}{n} \sum_{i=1}^{n} Z_i$. Therefore,

$$\mathbb{P}(Z - \mathbb{E}(Z) \leq -\tau) \leq \exp\left(\frac{-2n\tau^2}{(b-a)^2}\right).$$

# C Proof of Theorem 2.6

**Proof.** Denote $\tilde{\mathcal{W}} = \{\boldsymbol{w} \in \mathbb{R}_+^m \mid \tilde{\boldsymbol{B}}^T \boldsymbol{w} \leq \bar{d} \cdot \boldsymbol{e}\}$ and $\mathcal{S} = \{\boldsymbol{w} \in \mathbb{R}_+^m \mid \sum_{i=1}^{m} w_i \leq \bar{d}\}$. Our goal is to sandwich $\tilde{\mathcal{W}}$ between two simplicies and use Lemma 2.5. Using the following tail inequality for Gaussian random variables $\tilde{G} \sim \mathcal{N}(\mu, \sigma^2)$, $\mathbb{P}(|\tilde{G} - \mu| \geq t) \leq 2e^{-\frac{t^2}{2\sigma^2}}$, we have

$$\mathbb{P}(\tilde{B}_{ij} \leq \sqrt{6\log(mn)}) = 1 - \cdot\mathbb{P}\left(|\tilde{G}_{ij}| \geq \sqrt{6\log(mn)}\right)$$

$$\geq 1 - 2\exp\left(\frac{-6\log(mn)}{2}\right) = 1 - \frac{2}{(mn)^3} \geq 1 - \frac{1}{(mn)^2}$$

Therefore by taking a union bound,

$$\mathbb{P}\left(\tilde{B}_{ij} \leq \sqrt{6\log(mn)} \ \forall i \in [n], \forall j \in [m]\right) \geq \left(1 - \frac{1}{(mn)^2}\right)^{mn} \geq 1 - \frac{1}{mn}$$

where the last inequality follows from Bernoulli's inequality. Therefore for any $w \in \mathcal{S}$, we have with probability at least $1 - \frac{1}{mn}$,

$$\sum_{j=1}^{m} \tilde{B}_{ji} w_j \leq \sqrt{6\log(mn)} \sum_{j=1}^{m} w_j \leq \sqrt{6\log(mn)} \cdot \bar{d} \qquad \forall i \in [n]$$

Hence, with probability at least $1 - \frac{1}{mn}$ we have, $\mathcal{S} \subseteq \sqrt{6\log(mn)} \cdot \tilde{\mathcal{W}}$.

Now, we want to find a simplex that includes $\tilde{\mathcal{W}}$. We follow a similar approach to the proof of Theorem 2.1. Consider any $\boldsymbol{w} \in \tilde{\mathcal{W}}$. We have similarly to equation (7)

$$\sum_{j=1}^{m} \left(\frac{\sum_{i=1}^{n} \tilde{B}_{ji}}{n}\right) \cdot w_j \leq \bar{d}. \tag{12}$$

We have the following concentration inequality for non-negative random variables (see Theroem 7 in [12]),

$$\mathbb{P}\left(\frac{\sum_{i=1}^{n} \tilde{B}_{ji}}{n} \geq \mu - \tau\right) \geq 1 - \exp\left(\frac{-n\tau^2}{2\mathbb{E}(\tilde{B}_{11}^2)}\right) = 1 - \exp\left(\frac{-n\tau^2}{2}\right) = 1 - \frac{1}{m^2}$$

where $\tau = 2\sqrt{\frac{\log m}{n}}$ and $\mu = \mathbb{E}[\tilde{B}_{ji}] = \sqrt{\frac{2}{\pi}}$ is the expectation of a folded standard normal distribution. Then, union bound over $j = 1, \ldots, m$ gives us

$$\mathbb{P}\left(\frac{\sum_{i=1}^{n} \tilde{B}_{ji}}{n} \geq \mu - \tau \ \forall j = 1, \ldots, m\right) \geq \left(1 - \frac{1}{m^2}\right)^m \geq 1 - \frac{1}{m}.$$

where the last inequality follows from Bernoulli's inequality. Therefore, combining this result with inequality (12), we have with probability at least $1 - \frac{1}{m}$, $\tilde{\mathcal{W}} \subseteq \frac{1}{\mu - \tau}\mathcal{S}$. Denote, $\mathcal{S}' = \frac{1}{\sqrt{6\log(mn)}}\mathcal{S}$. Then, we have with probabilty at least $1 - \frac{1}{m}$, $\mathcal{S}' \subseteq \tilde{\mathcal{W}} \subseteq \kappa \cdot \mathcal{S}'$ where

$$\kappa = \frac{\sqrt{6\log(mn)}}{\sqrt{\frac{2}{\pi} - 2\sqrt{\frac{\log m}{n}}}} = O\left(\sqrt{\log m + \log n}\right),$$

for sufficiently large values of $m$ and $n$. We finally use Lemma 2.5 to conclude. □

# D  Proofs of Theorem 3.1

To prove Theorem 3.1, we introduce the following Lemma which shows a deterministic bad instance where the optimal affine solution is $\Theta(\sqrt{m})$ away from the optimal adjustable solution.

**Lemma D.1.** *Consider the two-stage adjustable problem* (1) *where:* $n = m, \boldsymbol{c} = \boldsymbol{0}, \ \boldsymbol{d} = \boldsymbol{e}, \boldsymbol{A} = \boldsymbol{0}$,

$$B_{ij} = \begin{cases} 1 & \text{if } i = j \\ \frac{1}{\sqrt{m}} & \text{if } i \neq j \end{cases} \tag{13}$$

*and the uncertainty set is defined as*

$$\mathcal{U} = \mathrm{conv}\left(\boldsymbol{0}, \boldsymbol{e}_1, \ldots, \boldsymbol{e}_m, \boldsymbol{\nu}_1, \ldots, \boldsymbol{\nu}_m\right) \tag{14}$$

*where* $\boldsymbol{\nu}_i = \frac{1}{\sqrt{m}}(\boldsymbol{e} - \boldsymbol{e}_i)$ *for* $i = 1, \ldots, m$. *Then,* $z_{\mathsf{Aff}}(\boldsymbol{B}) = \Omega(\sqrt{m}) \cdot z_{\mathsf{AR}}(\boldsymbol{B})$.

**Proof.** First, let us prove that $z_{\mathsf{AR}}(\boldsymbol{B}) \leq 1$. It is sufficient to define an adjustable solution only for the extreme points of $\mathcal{U}$ because the constraints are linear. We define the following solution for all $i = 1, \ldots, m$.

$$\boldsymbol{x} = \boldsymbol{0}, \qquad \boldsymbol{y}(\boldsymbol{0}) = \boldsymbol{0}, \qquad \boldsymbol{y}(\boldsymbol{e}_i) = \boldsymbol{e}_i, \qquad \boldsymbol{y}(\boldsymbol{\nu}_i) = \frac{1}{m}\boldsymbol{e}.$$

We have $\boldsymbol{B}\boldsymbol{y}(\boldsymbol{0}) = \boldsymbol{0}$ and for $i \in [m]$

$$\boldsymbol{B}\boldsymbol{y}(\boldsymbol{e}_i) = \boldsymbol{e}_i + \frac{1}{\sqrt{m}}(\boldsymbol{e} - \boldsymbol{e}_i) \geq \boldsymbol{e}_i$$

and

$$\boldsymbol{B}\boldsymbol{y}(\boldsymbol{\nu}_i) = \frac{1}{m}\boldsymbol{B}\boldsymbol{e} = \left(\frac{1}{m} + \frac{m-1}{m\sqrt{m}}\right)\boldsymbol{e} \geq \frac{1}{\sqrt{m}}\boldsymbol{e} \geq \boldsymbol{\nu}_i$$

Therefore, the solution defined above is feasible. Moreover, the cost of our feasible solution is 1 because for all $i \in [m]$, we have

$$\boldsymbol{d}^T\boldsymbol{y}(\boldsymbol{e}_i) = \boldsymbol{d}^T\boldsymbol{y}(\boldsymbol{\nu}_i) = 1.$$

Hence, $z_{\mathsf{AR}}(\boldsymbol{B}) \leq 1$. Now, it is sufficient to prove that $z_{\mathsf{Aff}}(\boldsymbol{B}) = \Omega(\sqrt{m})$. From Lemma 8 in Bertsimas and Goyal [8], since our instance is symmetric, i.e. $\mathcal{U}$ and $\mathcal{W}$ are permutation invariant, where $\mathcal{W}$ is the dualized uncertainty set, there exists an optimal solution for the affine problem (2) of the following form $\boldsymbol{y}(\boldsymbol{h}) = \boldsymbol{P}\boldsymbol{h} + \boldsymbol{q}$ for $\boldsymbol{h} \in \mathcal{U}$ where

$$\boldsymbol{P} = \begin{pmatrix} \theta & \mu & \cdots & \mu \\ \mu & \theta & \cdots & \mu \\ \vdots & \vdots & \ddots & \vdots \\ \mu & \mu & \cdots & \theta \end{pmatrix} \tag{15}$$

and $\boldsymbol{q} = \lambda\boldsymbol{e}$.

We have $\boldsymbol{y}(\boldsymbol{0}) = \lambda\boldsymbol{e} \geq \boldsymbol{0}$ hence

$$\lambda \geq 0. \tag{16}$$

We know that

$$z_{\mathsf{Aff}}(\boldsymbol{B}) \geq \boldsymbol{d}^T\boldsymbol{y}(\boldsymbol{0}) = \lambda m. \tag{17}$$

**Case 1:** If $\lambda \geq \frac{1}{6\sqrt{m}}$, then from (17) we have $z_{\mathsf{Aff}}(\boldsymbol{B}) \geq \frac{\sqrt{m}}{6}$.

**Case 2:** If $\lambda \leq \frac{1}{6\sqrt{m}}$. We have

$$\boldsymbol{y}(\boldsymbol{e}_1) = (\theta + \lambda)\boldsymbol{e}_1 + (\mu + \lambda)(\boldsymbol{e} - \boldsymbol{e}_1).$$

By feasibility of the solution, we have $\boldsymbol{B}\boldsymbol{y}(\boldsymbol{e}_1) \geq \boldsymbol{e}_1$, hence

$$(\theta + \lambda) + \frac{1}{\sqrt{m}}(m-1)(\mu + \lambda) \geq 1$$

Therefore $\theta + \lambda \geq \frac{1}{2}$ or $\frac{1}{\sqrt{m}}(m-1)(\mu + \lambda) \geq \frac{1}{2}$.

**Case 2.1:** Suppose $\frac{1}{\sqrt{m}}(m-1)(\mu + \lambda) \geq \frac{1}{2}$. Therefore,

$$z_{\text{Aff}}(\boldsymbol{B}) \geq \boldsymbol{d}^T \boldsymbol{y}(\boldsymbol{e}_1) = \theta + \lambda + (m-1)(\mu + \lambda) \geq \frac{\sqrt{m}}{2}.$$

where the last inequality holds because $\theta + \lambda \geq 0$ as $\boldsymbol{y}(\boldsymbol{e}_1) \geq \boldsymbol{0}$.

**Case 2.2:** Now suppose we have the other inequality i.e. $\theta + \lambda \geq \frac{1}{2}$. Recall that we have $\lambda \leq \frac{1}{6\sqrt{m}}$ as well. Therefore,

$$\theta \geq \frac{1}{2} - \frac{1}{6\sqrt{m}} \geq \frac{1}{3}.$$

We have,

$$\boldsymbol{y}(\boldsymbol{\nu}_1) = \frac{1}{\sqrt{m}}\left((\theta + (m-2)\mu)(\boldsymbol{e} - \boldsymbol{e}_1) + (m-1)\mu \boldsymbol{e}_1\right) + \lambda \boldsymbol{e}.$$

In particular we have ,

$$z_{\text{Aff}}(\boldsymbol{B}) \geq \boldsymbol{d}^T \boldsymbol{y}(\boldsymbol{\nu}_1) = \frac{1}{\sqrt{m}}((m-1)\theta + (m-1)^2\mu) + \lambda m$$

$$\geq \frac{m-1}{\sqrt{m}}\left(\frac{1}{3} + (m-1)\mu\right). \tag{18}$$

where the last inequality follows from $\lambda \geq 0$ and $\theta \geq \frac{1}{3}$.

**Case 2.2.1:** If $\mu \geq 0$ then from (18)

$$z_{\text{Aff}}(\boldsymbol{B}) \geq \frac{m-1}{3\sqrt{m}} = \Omega(\sqrt{m}).$$

**Case 2.2.2:** Now suppose that $\mu < 0$, by non-negativity of $\boldsymbol{y}(\boldsymbol{\nu}_1)$ we have

$$\frac{m-1}{\sqrt{m}}\mu + \lambda \geq 0$$

i.e.

$$\mu \geq \frac{-\lambda\sqrt{m}}{m-1}$$

and from (18)

$$z_{\text{Aff}}(\boldsymbol{B}) \geq \frac{m-1}{\sqrt{m}}\left(\frac{1}{3} + (m-1)\mu\right)$$

$$\geq \frac{m-1}{\sqrt{m}}\left(\frac{1}{3} - \lambda\sqrt{m}\right)$$

$$\geq \frac{m-1}{\sqrt{m}}\left(\frac{1}{3} - \frac{1}{6}\right) = \frac{m-1}{6\sqrt{m}} = \Omega(\sqrt{m}).$$

We conclude that in all cases $z_{\text{Aff}}(\boldsymbol{B}) = \Omega(\sqrt{m})$ and consequently $z_{\text{Aff}}(\boldsymbol{B}) = \Omega(\sqrt{m}) \cdot z_{\text{AR}}(\boldsymbol{B})$. $\qquad \square$

**Proof of Theorem 3.1**

**Proof.** Denote $\mathcal{W} = \{\boldsymbol{w} \in \mathbb{R}_+^m \mid \boldsymbol{B}^T\boldsymbol{w} \leq \bar{d}\boldsymbol{e}\}$ and $\tilde{\mathcal{W}} = \{\boldsymbol{w} \in \mathbb{R}_+^m \mid \tilde{\boldsymbol{B}}^T\boldsymbol{w} \leq \bar{d}\boldsymbol{e}\}$ where $\boldsymbol{B}$ is defined in (13) and $\tilde{\boldsymbol{B}}$ is defined in (8). Since for all $i, j$ in $\{1, \ldots, m\}$ we have $\tilde{B}_{ij} \leq B_{ij}$. Hence, for any $\boldsymbol{w} \in \mathcal{W}$, we have $\tilde{\boldsymbol{B}}^T\boldsymbol{w} \leq \boldsymbol{B}^T\boldsymbol{w} \leq \bar{d}\boldsymbol{e}$. Therefore $\boldsymbol{w} \in \tilde{\mathcal{W}}$ and consequently $\mathcal{W} \subseteq \tilde{\mathcal{W}}$.

Now, suppose $\boldsymbol{w} \in \tilde{\mathcal{W}}$, we have for all $i = 1, \ldots, m$

$$w_i + \frac{1}{\sqrt{m}}\sum_{\substack{j=1 \\ j \neq i}}^{m} \tilde{u}_{ji} w_j \leq \bar{d}. \tag{19}$$

By taking the sum over $i$, dividing by $m$ and rearranging, we get

$$\sum_{i=1}^{m} w_i \left(\frac{1}{m} + \frac{1}{m\sqrt{m}}\sum_{\substack{j=1 \\ j \neq i}}^{m} \tilde{u}_{ij}\right) \leq \bar{d}. \tag{20}$$

Here, similarly to the proof of Lemma 2.1 we apply Hoeffding's inequality [18](see appendix B), with $\tau = \sqrt{\frac{\log m}{m-1}}$

$$\mathbb{P}\left(\frac{\sum_{j\neq i}^m \tilde{u}_{ij}}{m-1} \geq \frac{1}{2} - \tau\right) \geq 1 - \exp\left(-2(m-1)\tau^2\right) = 1 - \frac{1}{m^2}$$

and we take a union bound over $j = 1, \ldots, m$

$$\mathbb{P}\left(\frac{\sum_{i=1}^n \tilde{u}_{ij}}{m-1} \geq \frac{1}{2} - \epsilon \ \ \forall j = 1, \ldots, m\right) \geq \left(1 - \frac{1}{m^2}\right)^m \geq 1 - \frac{1}{m}. \tag{21}$$

where the last inequality follows from Bernoulli's inequality. Therefore, we conclude from (20) and (21), that with probability at least $1 - \frac{1}{m}$ we have $\beta \sum_{i=1}^m w_i \leq \bar{d}$ where $\beta = \frac{1}{m} + \frac{m-1}{m\sqrt{m}}(\frac{1}{2} - \tau) \geq \frac{1}{4\sqrt{m}}$ for $m$ sufficiently large. Note from (19) that for all $i$ we have $w_i \leq \bar{d}$. Hence with probability at least $1 - \frac{1}{m}$, we have for all $i = 1, \ldots, m$

$$\boldsymbol{B}_i^T w = w_i + \frac{1}{\sqrt{m}} \sum_{\substack{j=1 \\ j \neq i}}^m w_j \leq \bar{d} + \frac{\bar{d}}{\beta\sqrt{m}} \leq 5 \cdot \bar{d}$$

Therefore, $\boldsymbol{w} \in 5 \cdot \mathcal{W}$ for any $\boldsymbol{w}$ in $\mathcal{W}$ and consequently we have with probability at least $1 - \frac{1}{m}$, $\tilde{\mathcal{W}} \subseteq 5 \cdot \mathcal{W}$. All together we have proved with probability at least $1 - \frac{1}{m}$ $\mathcal{W} \subseteq \tilde{\mathcal{W}} \subseteq 5 \cdot \mathcal{W}$. This implies with probability at least $1 - \frac{1}{m}$, that $z_{\mathsf{d-Aff}}(\tilde{\boldsymbol{B}}) \geq z_{\mathsf{d-Aff}}(\boldsymbol{B})$ and $z_{\mathsf{d-AR}}(\boldsymbol{B}) \geq \frac{z_{\mathsf{d-AR}}(\tilde{\boldsymbol{B}})}{5}$. We know from from Lemma 2.3 and Lemma 2.2 that the dualized and primal are the same both for the adjustable problem and affine problem. Hence, with probability at least $1 - \frac{1}{m}$, we have $z_{\mathsf{Aff}}(\tilde{\boldsymbol{B}}) \geq z_{\mathsf{Aff}}(\boldsymbol{B})$ and $z_{\mathsf{AR}}(\boldsymbol{B}) \geq \frac{z_{\mathsf{AR}}(\tilde{\boldsymbol{B}})}{5}$.

Moreover, we know from Lemma D.1 that $z_{\mathsf{Aff}}(\boldsymbol{B}) \geq \Omega(\sqrt{m}) \cdot z_{\mathsf{AR}}(\boldsymbol{B})$. Therefore, $z_{\mathsf{Aff}}(\tilde{\boldsymbol{B}}) \geq \Omega(\sqrt{m}) z_{\mathsf{AR}}(\tilde{\boldsymbol{B}})$ with probability at least $1 - \frac{1}{m}$.

$\square$

# E LP and MIP formulations for the empirical section

**LP formulation for the affine problem.** The affine problem (2) can be formulated as the following LP

$$
\begin{aligned}
z_{\mathsf{Aff}}(\boldsymbol{B}) = \min \ & \boldsymbol{c}^T \boldsymbol{x} + z \\
& z - \boldsymbol{d}^T \boldsymbol{q} \geq \boldsymbol{r}^T \boldsymbol{v} \\
& \boldsymbol{R}^T \boldsymbol{v} \geq \boldsymbol{P}^T \boldsymbol{d} \\
& \boldsymbol{A}\boldsymbol{x} + \boldsymbol{B}\boldsymbol{q} \geq \boldsymbol{V}^T \boldsymbol{r} \\
& \boldsymbol{R}^T \boldsymbol{V} \geq \boldsymbol{I}_m - \boldsymbol{B}\boldsymbol{P} \\
& \boldsymbol{q} \geq \boldsymbol{U}^T \boldsymbol{r} \\
& \boldsymbol{U}^T \boldsymbol{R} + \boldsymbol{P} \geq \boldsymbol{0} \\
& \boldsymbol{x} \in \mathbb{R}_+^n, \ \boldsymbol{v} \in \mathbb{R}_+^L, \ \boldsymbol{U} \in \mathbb{R}_+^{L \times n}, \ \boldsymbol{V} \in \mathbb{R}_+^{L \times m}
\end{aligned}
\tag{22}
$$

**Proof.** The affine problem (2) can be reformulated as follows

$$
\begin{aligned}
z_{\mathsf{Aff}}(\boldsymbol{B}) = \min_{\boldsymbol{x}} \ & \boldsymbol{c}^T \boldsymbol{x} + z \\
& z \geq \boldsymbol{d}^T \left(\boldsymbol{P}\boldsymbol{h} + \boldsymbol{q}\right) \quad \forall \boldsymbol{h} \in \mathcal{U} \\
& \boldsymbol{A}\boldsymbol{x} + \boldsymbol{B}\left(\boldsymbol{P}\boldsymbol{h} + \boldsymbol{q}\right) \geq \boldsymbol{h} \quad \forall \boldsymbol{h} \in \mathcal{U} \\
& \boldsymbol{P}\boldsymbol{h} + \boldsymbol{q} \geq \boldsymbol{0} \quad \forall \boldsymbol{h} \in \mathcal{U} \\
& \boldsymbol{x} \in \mathbb{R}_+^n
\end{aligned}
$$

We use standard duality techniques to derive formulation (22). The first constraint is equivalent to

$$z - \boldsymbol{d}^T \boldsymbol{q} \geq \max_{\substack{\boldsymbol{R}\boldsymbol{h} \leq \boldsymbol{r} \\ \boldsymbol{h} \geq \boldsymbol{0}}} \boldsymbol{d}^T \boldsymbol{P}\boldsymbol{h}.$$

By taking the dual of the maximization problem, the constraint is equivalent

$$z - \boldsymbol{d}^T \boldsymbol{q} \geq \min_{\substack{\boldsymbol{R}^T \boldsymbol{v} \geq \boldsymbol{P}^T \boldsymbol{d} \\ \boldsymbol{v} \geq \boldsymbol{0}}} \boldsymbol{r}^T \boldsymbol{v}$$

We can then drop the min and introduce $\boldsymbol{v}$ as a variable, hence we obtain the following linear constraints

$$z - \boldsymbol{d}^T\boldsymbol{q} \geq \boldsymbol{r}^T\boldsymbol{v}$$
$$\boldsymbol{R}^T\boldsymbol{v} \geq \boldsymbol{P}^T\boldsymbol{d}$$
$$\boldsymbol{v} \in \mathbb{R}^L_+$$

We use the same technique for the second sets of constraints, i.e.

$$\boldsymbol{Ax} + \boldsymbol{Bq} \;\geq\; \max_{\substack{\boldsymbol{Rh}\leq\boldsymbol{r} \\ \boldsymbol{h}\geq\boldsymbol{0}}} \boldsymbol{h}(\boldsymbol{I}_m - \boldsymbol{BP})$$

By taking the dual of the maximization problem for each row and dropping the min we get the following compact formulation of these constraints

$$\boldsymbol{Ax} + \boldsymbol{Bq} \geq \boldsymbol{V}^T\boldsymbol{r}$$
$$\boldsymbol{R}^T\boldsymbol{V} \geq \boldsymbol{I}_m - \boldsymbol{BP}$$
$$\boldsymbol{V} \in \mathbb{R}^{L\times m}_+$$

Similarly, the last constraint

$$\boldsymbol{q} \;\geq\; \max_{\substack{\boldsymbol{Rh}\leq\boldsymbol{r} \\ \boldsymbol{h}\geq\boldsymbol{0}}} \;- \boldsymbol{Ph}$$

is equivalent to

$$\boldsymbol{q} \geq \boldsymbol{U}^T\boldsymbol{r}$$
$$\boldsymbol{U}^T\boldsymbol{R} + \boldsymbol{P} \geq \boldsymbol{0}$$
$$\boldsymbol{U} \in \mathbb{R}^{L\times n}_+.$$

$\square$

**MIP formulation for the separation adjustable problem.**

The separation problem (10) can be formulated as the following MIP

$$
\begin{aligned}
\max \quad & \sum_{i=1}^{m}\sum_{j=-\Delta_{\mathcal{W}}}^{s}\sum_{k=-\Delta_{\mathcal{U}}}^{s} \frac{1}{2^{j+k}}\cdot\gamma_{ijk} - (\boldsymbol{A\hat{x}})^T\boldsymbol{w} \\
& \boldsymbol{w} = \sum_{i=1}^{m}\sum_{j=-\Delta_{\mathcal{W}}}^{s} \frac{\beta_{ij}}{2^j}\cdot\boldsymbol{e}_i \\
& \boldsymbol{h} = \sum_{i=1}^{m}\sum_{k=-\Delta_{\mathcal{U}}}^{s} \frac{\alpha_{ik}}{2^k}\cdot\boldsymbol{e}_i \\
& \gamma_{ijk} \leq \beta_{ij} && \forall i\in[m], j\in[-\Delta_{\mathcal{U}},s], k\in[-\Delta_{\mathcal{W}},s] \\
& \gamma_{ijk} \leq \alpha_{ik} && \forall i\in[m], j\in[-\Delta_{\mathcal{U}},s], k\in[-\Delta_{\mathcal{W}},s] \\
& \gamma_{ijk} + 1 \geq \alpha_{ik} + \beta_{ij} && \forall i\in[m], j\in[-\Delta_{\mathcal{U}},s], k\in[-\Delta_{\mathcal{W}},s] \\
& \alpha_{ik}, \beta_{ik}, \gamma_{ijk} \in \{0,1\} && \forall i\in[m], j\in[-\Delta_{\mathcal{U}},s], k\in[-\Delta_{\mathcal{W}},s] \\
& \boldsymbol{Rh} \leq \boldsymbol{r} \\
& \boldsymbol{B}^T\boldsymbol{w} \leq \boldsymbol{d}
\end{aligned}
\tag{23}
$$

where $s = \lceil \log_2\left(\frac{m}{\epsilon}\right)\rceil$, $\Delta_{\mathcal{W}}$ is an upper bound on any component of $w\in\mathcal{W}$, $\Delta_{\mathcal{U}}$ is an upper bound on any component of $h\in\mathcal{U}$ and $\epsilon$ is the accuracy of the problem.

**Proof.** The separation problem (10) is equivalent to solving the following problem for given $\hat{\boldsymbol{x}}$

$$\max_{\substack{\boldsymbol{h}\in\mathcal{U} \\ \boldsymbol{w}\in\mathcal{W}}} \boldsymbol{h}^T\boldsymbol{w} - (\boldsymbol{A\hat{x}})^T\boldsymbol{w}$$

The constraints of the above problem are linear and the second term in the objective function is linear as well. So we will focus only on the first term $\boldsymbol{h}^T\boldsymbol{w}$ which is a bilinear function and write it in terms of linear constraints and binary variables. Let us write $\boldsymbol{h} = \sum_{i=1}^{m} h_i\boldsymbol{e}_i$. For all $i\in[m]$ we digitize the component $h_i$ as follows

$$h_i = \sum_{k=-\Delta_{\mathcal{U}}}^{s} \frac{\alpha_{ik}}{2^k}$$

where $s = \lceil \log_2 \left( \frac{m}{\epsilon} \right) \rceil$, $\Delta_{\mathcal{U}}$ is an upper bound on any $h_i$ and $\alpha_{ik}$ are binary variables. This digitization gives an approximation to $h_i$ within $\frac{\epsilon}{m}$ which translates to an accuracy of $\epsilon$ in the objective function. We have

$$\boldsymbol{h} = \sum_{i=1}^{m} \sum_{k=-\Delta_{\mathcal{U}}}^{s} \frac{\alpha_{ik}}{2^k} \cdot \boldsymbol{e}_i$$

Similarly, we have

$$\boldsymbol{w} = \sum_{i=1}^{m} \sum_{j=-\Delta_{\mathcal{W}}}^{s} \frac{\beta_{ij}}{2^j} \cdot \boldsymbol{e}_i$$

where $\Delta_{\mathcal{W}}$ is an upper bound on any component of $w \in \mathcal{W}$. Therefore, the first term in the objective function becomes

$$\sum_{i=1}^{m} \sum_{j=-\Delta_{\mathcal{W}}}^{s} \sum_{k=-\Delta_{\mathcal{U}}}^{s} \frac{1}{2^{j+k}} \cdot \alpha_{ik} \beta_{ij}$$

The final step is to linearize the term $\alpha_{ik} \beta_{ij}$. We set, $\alpha_{ik} \beta_{ij} = \gamma_{ijk}$ where again $\gamma_{ijk}$ is a binary variable. Since all the variables here are binary we can express $\gamma_{ijk}$ using only linear constraints as follows

$$\gamma_{ijk} \leq \beta_{ij}$$

$$\gamma_{ijk} \leq \alpha_{ik}$$
$$\gamma_{ijk} + 1 \geq \alpha_{ik} + \beta_{ij}$$

which leads to formulation (23).

$\square$