[Reviews · NeurIPS 2017]

Reviewer 1



This paper studies the contrast between the worst-case and the empirical performance of affine policies. The authors show the affine policies produce a good approximation for the two-stage adjustable robust optimization problem with high probability on random instances where the constraint coefficients are generated i.i.d. from a large class of distribution. The empirical results affine policy is very close to optimal and also several magnitudes faster. In Page 3, the paper mentions MIP, but doesn’t define it. I think MIP is mixed integer program, which is mentioned in page 8. The paper misses its reference page (it is in the paper’s supplementary file).

Reviewer 2



The authors present an interesting analysis of a particular robust optimization problem: two stage adjustable robust linear optimization with an affine policy. Specifically, the authors explore the discrepancy between poor theoratical performance (according to worst-case theoretical bounds) and near-optimal observed empirical performance. Instead of looking at the worst case scenarios, the authors derive bounds on performance when the contraint matrix is generated from a particular class of probability dstitribution. They derive bounds for several families and iii/non iid generation scenarios. I am not particular familiar with this problem but the paper is written well, and the material is presented in a clear and accessible fashion. My only comment would be to add an example of a specific application ( and motivation) of the two-stage adjustable robust linear optimization. Readers who are not familiar with the subject matter will have greater appreciation for the presented results.

Reviewer 3



Review 2 after authors' comments: I believe the authors gave a very good rebuttal to the comments made which leads me to believe that there is no problem in accepting this paper, see updated rating. ------------ This paper addresses the challenging question of giving bounds for affine policies for adjustable robust optimization models. I like the fact that the authors (in some probabilistic sense) reduced the large gap of sqrt(m) to 2 for some specific instances. The authors have combined various different techniques combining probabilistic bounds to the structure of a set, which is itself derived using a dual-formulation from the original problem. However, I believe the contributions given in this paper and the impact is not high enough to allow for acceptance for NIPS. I came to this conclusion due to the following reasons: - Bounds on the performance of affine policies have been described in a series of earlier papers. This paper does not significantly close the gap in my opinion. - The results strongly depend on the generative model for the instances. However, adjustable robust optimization is solely used in environments that have highly structured models, such as the ones mentioned in the paper on page 2, line 51: set cover, facility location and network design problems. It is also explained that the performance differs if the generative model, or the i.i.d. assumption is slightly changed. Therefore, I am not convinced about the insights these results give for researchers that are thinking of applying adjustable robust optimization to solve their (structured) problem. - Empirical results are much better than the bounds presented here. In particular, it appears that affine policies are near optimal. This was known and has been shown in various other papers before. - On page 2, lines 48-49 the authors say that "...without loss of generality that c=e and d=\bar{d}e (by appropriately scaling A and B)." However, I believe you also have to scale the right-hand side h (or the size/shape of the uncertainty set). And the columns of B have to be scaled, making the entries no longer I.I.D. in the distribution required in Section 2? - Also on page 2, line 69 the authors describe the model with affine policies. The variables P and q are still in an inner minimization model. I think they should be together with the minimization over x? - On page 6, Theorem 2.6. The inequality z_AR <= z_AFF does always hold I believe? So not only with probability 1-1/m? - There are a posteriori methods to describe the optimality gap of affine policies that are much tighter for many applications, such as the methods described by [Hadjiyiannis, Michael J., Paul J. Goulart, and Daniel Kuhn. "A scenario approach for estimating the suboptimality of linear decision rules in two-stage robust optimization." Decision and Control and European Control Conference (CDC-ECC), 2011 50th IEEE Conference on. IEEE, 2011.] and [Kuhn, Daniel, Wolfram Wiesemann, and Angelos Georghiou. "Primal and dual linear decision rules in stochastic and robust optimization." Mathematical Programming 130.1 (2011): 177-209.]. ------- I was positively surprised by the digitized formulation on page 8. Is this approach used before in the literature, and if so where? The authors describe that the size can depend on desired accuracy. With the given accuracy, is the resulting solution a lower bound or an upper bound? If it is an upper bound, is the solution feasible? Of course, because it is a MIP it can probably only solve small problems as also illustrated by the authors. ------- I have some minor comments concerning the very few types found on page 5 (this might not even be worth mentioning here): - line 166. "peliminary" - line 184. "affinly independant" - line 186. "The proof proceeds along similar lines as in 2.4." (<-- what does 2.4 refer to?)